# Is It Possible to Be Happy during the COVID-19 Lockdown? A Longitudinal Study of the Role of Emotional Regulation Strategies and Pleasant Activities in Happiness

**DOI:** 10.3390/ijerph18063211

**Published:** 2021-03-19

**Authors:** María José Gutiérrez-Cobo, Alberto Megías-Robles, Raquel Gómez-Leal, Rosario Cabello, Pablo Fernández-Berrocal

**Affiliations:** 1Department of Developmental and Educational Psychology, Faculty of Psychology, University of Málaga, 29071 Málaga, Spain; 2Department of Basic Psychology, Faculty of Psychology, University of Málaga, 29071 Málaga, Spain; amegias@uma.es (A.M.-R.); raqgomlea@uma.es (R.G.-L.); berrocal@uma.es (P.F.-B.); 3Department of Developmental and Educational Psychology, Faculty of Psychology, University of Granada, 18071 Granada, Spain; rcabello@ugr.es

**Keywords:** cognitive reappraisal, expressive suppression, pleasant activities, happiness, COVID-19

## Abstract

This study aimed to longitudinally analyze the role played by two emotional regulation strategies (cognitive reappraisal and expressive suppression), through the mediating effect of engagement in pleasant activities during lockdown, in changes in affective and cognitive happiness in comparison with pre-pandemic levels. Eighty-eight participants from a community sample were evaluated at two timepoints. At timepoint 1 (before the COVID-19 pandemic), participants were evaluated on emotional regulation and cognitive and affective happiness. At timepoint 2 (during the COVID-19 lockdown), participants were evaluated on cognitive and affective happiness and the frequency with which they engaged in pleasant activities. We found an optimal fit of the proposed model in which cognitive reappraisal was significantly related to engagement in more pleasant activities during the lockdown. In turn, these pleasant activities were related to more affective happiness during the lockdown (compared with pre-pandemic levels), and this affective happiness was associated with greater cognitive happiness. In conclusion, cognitive reappraisal was a protective factor for affective and cognitive happiness through the mediating role of engagement in pleasant activities during lockdown. Limitations and future lines of investigation are discussed.

## 1. Introduction

Since the worldwide emergence of COVID-19, an extensive body of research has been conducted to clarify the impact of the pandemic on mental health and well-being [1,2,3,4,5]. The results of all these studies seem to indicate that there has been an overall decline in the mental health of the general population, with high percentages of depression, anxiety, and stress, particularly in risk populations. Of particular relevance are the consequences of the special measures that the authorities put in place during the lockdown periods where the freedom of citizens was severely restricted [6,7].

The previous literature on the mental health consequences of COVID-19 has mainly employed cross-sectional methodologies. These methodologies, however, do not allow for capturing changes in the psychological impact and predictors of the COVID-19 lockdown. Therefore, longitudinal methodologies should be used to address this gap in order to identify the real changes that have occurred during the pandemic. In addition, previous studies have focused on psychopathology, with rather less attention being paid to positive psychology and the protective factors of people’s happiness during the COVID-19 lockdown. Happiness is understood as a balance between pleasant and unpleasant experiences, with many of the former and relatively few of the latter [8]. A happy life is dependent on an affective and cognitive component [9], with these components being interrelated [10]. Following the time-sequential model of well-being [11], the affective component has a significant impact on the cognitive component, given that individuals use the former to judge life satisfaction. To evaluate the affective component, individuals judge their positive and negative affects whilst analysis of the cognitive component involves a subjective evaluation of their life in general. Leading a happy life has been related to positive outcomes such as longevity or good physical and mental health [12,13,14].

One of the central aims of positive psychology has been to discover how to improve happiness. Following the Sustainable Happiness Model [15], happiness is influenced by the following three components: genetics, life circumstances, and intentional activities. Even though some of these components leave little space for potential modification, individuals can play a role in improving their happiness levels by managing their intentional activities. Such activities could account for approximately 15% of the variance in our happiness [16]. The COVID-19 confinement period in Spain (15 March to 4 May 2020) was an uncontrollable situation and one of the strictest lockdowns worldwide. Nonetheless, individuals may have intentionally acted to protect their happiness by frequently carrying out pleasant activities such as exercising at home, reading, and watching TV.

Individual differences in the frequency with which these pleasant activities are carried out have previously been linked to emotional regulation strategies. Emotional regulation refers to the individual efforts to influence emotions [17]. There are a variety of emotional regulation strategies, with expressive suppression and cognitive reappraisal being the most widely studied. Following the model of emotional regulation [18], cognitive reappraisal implies a cognitive change in the reinterpretation of a situation or goal (e.g., not interpreting the lockdown as a punishment, but as a way of avoiding the spread of the disease to both ourselves and our loved ones) and it is regarded as an adaptative strategy. Expressive suppression implies a response modulation where the individual avoids the external display of an internal emotional state (e.g., avoiding the expression of the anxiety caused by the lockdown). Cognitive reappraisal, unlike expressive suppression, has previously been linked to better mental health (e.g., fewer depressive symptoms) and—of particular relevance for the purposes of the present study—to happier individuals [17,18,19,20,21,22,23,24]. In this regard, those individuals employing cognitive reappraisal seem to engage in more pleasant activities such as practicing sport regularly [25]. In contrast, expressive suppression has a negative impact on emotions and can thus be regarded as a risk factor for happiness [20].

The main objective of the present study was to analyze the potential role of two emotional regulation strategies (cognitive reappraisal and expressive suppression) as protective or risk factors for alterations in affective and cognitive happiness through the mediating role of engagement in pleasant activities during lockdown. We employed a longitudinal methodology by analyzing affective and cognitive happiness at two timepoints: before the COVID-19 outbreak, and during the lockdown in Spain, where very strict confinement measures were applied from 15 March to 4 May 2020. The COVID-19 pandemic has changed the way we interact with our environment in the short, medium, and long-term [20]. There is still a high degree of uncertainty about how this situation will evolve, which makes it necessary to focus on the psychological mechanisms that will make it easier for individuals to deal with this crisis. In this regard, the present longitudinal study could offer relevant information for facing situations of restricted freedom in an optimal way as well as elucidating the mechanisms that could potentiate affective and cognitive happiness.

On the basis of results reported in the previous literature, we proposed the following hypotheses:

**Hypothesis** **1** **(H1).***Engaging in pleasant activities is positively correlated with affective happiness*.

**Hypothesis** **2** **(H2).***Engaging in pleasant activities is positively correlated with cognitive happiness*.

**Hypothesis** **3** **(H3).***Cognitive reappraisal is positively correlated with engaging in pleasant activities*.

**Hypothesis** **4** **(H4).***Cognitive reappraisal is positively correlated with affective happiness*.

**Hypothesis** **5** **(H5).***Cognitive reappraisal is positively correlated with cognitive happiness*.

**Hypothesis** **6** **(H6).***Expressive suppression is negatively correlated with engaging in pleasant activities*.

**Hypothesis** **7** **(H7).***Expressive suppression is negatively correlated with affective happiness*.

**Hypothesis** **8** **(H8).***Expressive suppression is negatively correlated with cognitive happiness*.

**Hypothesis** **9** **(H9).***Engaging in pleasant activities mediates the relationship between emotional regulation strategies and affective and cognitive happiness*.

## 2. Materials and Methods

### 2.1. Participants

Eighty-eight participants from a community sample voluntarily took part in this study. Their ages ranged from 19 to 67 years (M = 29.09; SD = 12.77) and 59 were women (67.05%). The participants were recruited by snowball sampling with the help of undergraduate students from the University of Málaga, Spain. Participants were assured about the confidentiality and anonymity of the collected data, and they were treated in accordance with the Helsinki declaration [26].

### 2.2. Instruments

The Emotion Regulation Questionnaire (ERQ) [22]. This instrument is a 10-item self-report questionnaire developed to assess two emotion regulation strategies based on Gross’s model of emotion regulation [27]: cognitive reappraisal and expressive suppression. Four items are related to expressive suppression and six items to cognitive reappraisal. Participants must indicate the extent to which they agree with each item statement on a 7-point Likert scale (ranging from 1 “strongly disagree” to 7 “strongly agree”). In the present study, the Spanish version of the ERQ [22] was used, which has been shown to have adequate internal consistency (Cronbach’s α = 0.75 for expressive suppression, Cronbach’s α = 0.79 for cognitive reappraisal). The internal consistency for the sample of this study was also adequate (Cronbach’s α = 0.78 for expressive suppression, Cronbach’s α = 0.79 for cognitive reappraisal).

The Positive and Negative Affect Schedule (PANAS) [28]. This instrument was employed as a measure of affective happiness [29], following previous studies. This is a widely used and well-validated self-report scale for measuring both positive affect (PA) and negative affect (NA). The 10-item short form version of the scale was used [30], in which five items assess PA and another five items assess NA. Participants must respond to each item on a 5-point Likert scale ranging from 1 “not at all” to 5 “strongly” where they must indicate the extent to which they have felt a particular emotion during the two previous weeks. In the current study, the measure of affective happiness was the affect balance score [31], computed as the difference between mean PA score and mean NA score (i.e., PA–NA). The internal consistency for the sample of this study was acceptable (timepoint 1: Cronbach’s α = 0.72 for PA; Cronbach’s α = 0.71 for NA; timepoint 2: Cronbach’s α = 0.73 for PA; Cronbach’s α = 0.74 for NA).

Cognitive happiness was assessed using one of the items of the mental health subscale of the SF-36 health survey [32] (adapted to Spanish by Alonso, Prieto, and Anto [33]). This item was as follows: “During the past two weeks, how often have you felt happy?” (In Spanish: “Durante las últimas dos semanas ¿con que frecuencia te sentiste feliz?”). Participants had to respond on a 6-point Likert scale ranging from 1 “none of the time” to 6 “all of the time”.

The frequency with which pleasant activities were carried out during the lockdown was assessed by the following statement: “In the last two weeks, I have done activities at home that I like and that make me feel good. For example, exercise, reading, watching TV shows, listening to music, cooking, etc.” (In Spanish: “En estas dos últimas semanas he hecho actividades en casa que me gustan y me hacen sentir bien. Por ejemplo: ejercicio, ver series, leer, escuchar música, cocinar, etc.”). Participants responded on a 7-point Likert-scale in which 1 indicates “totally disagree” and 7 indicates “totally agree”.

### 2.3. Procedure

The study consisted of two assessment phases. A first phase (hereinafter timepoint 1), which was conducted before the COVID-19 outbreak (11–26 November 2019), and a second phase (hereinafter timepoint 2), which was conducted four months later during the Spanish lockdown imposed in response to the COVID-19 pandemic (1–16 April 2020).

The assessment at timepoint 1 was part of a larger project concerned with emotional abilities and well-being (project UMA18-FEDERJA-114). In this first assessment, participants completed a battery of questionnaires that included the ERQ, the PANAS, the item on cognitive happiness belonging to the SF-36 health survey, and questions on demographic characteristics. Seven hundred and eighty-three participants completed the questionnaires in this first phase. These participants were recruited by snowball sampling and completed the questionnaires through online platforms (LimeSurvey and Google forms). To avoid missing data, blank responses were not allowed in the questionnaires.

For the assessment carried out at timepoint 2, we invited all the people who had completed the first assessment to participate in this second phase. In this second assessment, participants had to complete again the PANAS and the item on cognitive happiness, in order to explore possible changes between timepoint 1 and 2. Moreover, they had to respond to a set of questions on their experience during the COVID-19 lockdown, among which was our question of interest about the frequency with which they engaged in pleasant activities. For participating in this second assessment phase, the respondents were compensated with the chance to win a €40 Amazon gift card. A total number of 88 participants completed these questionnaires, which represents the final sample of the study that completed both the timepoint 1 and timepoint 2 assessment. As in the assessment at timepoint 1, all the questionnaires were administered online, and blank responses were not allowed in order to avoid missing data.

### 2.4. Statistical Analysis

Before conducting the analyses, we computed differential scores for affective happiness (from PANAS) and for cognitive happiness by subtracting the scores obtained before the COVID-19 outbreak from those obtained during the COVID-19 lockdown (hereinafter, affective happiness change and cognitive happiness change).

First, descriptive analyses (means and standard deviations) were conducted to explore the scores obtained for each measure. Second, gender and age-related differences in the study variables were examined using independent two-sample *t*-tests and linear and quadratic regression analysis, respectively. Moreover, changes in levels of affective and cognitive happiness between timepoint 1 and timepoint 2 were analyzed by paired sample *t*-tests. Third, the relationships between the variables included in the study were evaluated by bivariate Pearson’s correlations. The statistical significance threshold for *t*-tests, regression analyses, and Pearson’s correlations was set at *p* < 0.05. Fourth, and as the main objective of this research, a path analysis was conducted to evaluate the proposed model integrating emotion regulation strategies, engagement in pleasant activities during the COVID-19 lockdown, and changes in the levels of affective happiness and cognitive happiness during lockdown in comparison with pre-pandemic levels. Maximum likelihood estimation was used to estimate the model parameters, and the model fit to the data was tested by the following indices: χ^2^ statistic, comparative fit index (CFI), and the root mean square error of approximation (RMSEA) [34,35]. Indirect effects were computed with the bias-corrected bootstrapping method (5000 samples) and the level of statistical significance was set at 95% confidence intervals (CIs). All analyses were conducted using IBM SPSS 23 (IBM corp., Armonk, NY, USA) and IBM AMOS 21.0 (IBM corp., Armonk, NY, USA) software.

## 3. Results

Descriptive statistics for both the total sample and separated by gender are presented in Table 1. Gender differences were not observed for any variable (all *p*s > 0.05). Age was only found to be linearly related to the variable of affective happiness at timepoint 1 (β = 0.25, *p* = 0.02). The older the age, the higher the level of affective happiness at timepoint 1. Paired sample *t*-tests comparing the levels of cognitive happiness between timepoint 1 and timepoint 2 revealed higher levels of cognitive happiness at timepoint 1 (*t* (87) = 3.93; *p* < 0.001; Cohen’s *d* = 0.42). Differences in affective happiness between timepoint 1 and timepoint 2 were not significant (*p* > 0.05).

Pearson’s correlation analyses are presented in Table 2. Focusing on the correlations of interest, we observed that cognitive reappraisal was positively correlated with engagement in pleasant activities (*r* = 0.35, *p* < 0.001), affective happiness at timepoint 1 (*r* = 0.41, *p* < 0.001) and timepoint 2 (*r* = 0.35, *p* < 0.001), and cognitive happiness at timepoint 1 (*r* = 0.42, *p* < 0.001) and timepoint 2 (*r* = 0.27, *p* = 0.01). Expressive suppression was negatively correlated with affective happiness at timepoint 1 (*r* = −0.25, *p* = 0.02) and timepoint 2 (*r* = −0.24, *p* = 0.02). It is worth noting that, although non-significant, expressive suppression showed a correlation near to the statistical significance threshold with engagement in pleasant activities (*r* = −0.19, *p* = 0.07) and cognitive happiness at timepoint 1 (*r* = −0.19, *p* = 0.07) and timepoint 2 (*r* = −0.19, *p* = 0.07). With respect to engagement in pleasant activities, in addition to the relationships previously described, this variable showed a positive correlation with affective happiness at timepoint 1 (*r* = 0.28, *p* < 0.01) and timepoint 2 (*r* = 0.58, *p* < 0.001), affective happiness change (*r* = 0.32, *p* < 0.01), and cognitive happiness at timepoint 1 (*r* = 0.40, *p* < 0.001) and timepoint 2 (*r* = 0.51, *p* < 0.001). Finally, a positive correlation was observed between the affective happiness change and cognitive happiness change (*r* = 0.29, *p* < 0.01).

The results of the path analysis revealed an optimal fit of the proposed model (χ^2^(5) = 4.92, *p* = 0.42; CFI = 1.00; RMSEA < 0.001). Figure 1 shows the model with the standardized path coefficients and proportion of explained variance. According to these results, cognitive reappraisal was significantly associated with engagement in pleasant activities during the COVID-19 lockdown period. In turn, these pleasant activities were related to more positive affective happiness during the COVID-19 lockdown (in comparison with pre-pandemic levels), which was related to higher levels of cognitive happiness. The emotion regulation strategy of expressive suppression was not significantly related to engagement in pleasant activities; however, the resulting model after removing the path between expressive suppression and pleasant activities from the original model showed slightly worse fit indices (χ^2^(3) = 4.84, *p* = 0.18; CFI = 0.93; RMSEA = 0.08), although differences between models were not significant (χ^2^ diff = 1.87, df diff = 1, *p* > 0.05).

Finally, analysis of the indirect effects, as tested by the bootstrapping method (see Table 3) revealed that cognitive reappraisal was positively related to the cognitive happiness change through the effect on engagement in pleasant activities and the affective happiness change (standardized indirect effect = 0.030; 95% CI: 0.004, 0.080). Engagement in pleasant activities was also positively indirectly related to the cognitive happiness change through the affective happiness change (standardized indirect effect = 0.092; 95% CI: 0.011, 0.206). Expressive suppression did not show an indirect effect on the cognitive happiness change via engagement in pleasant activities and the affective happiness change (standardized indirect effect = −0.013; 95% CI: −0.057, 0.002).

## 4. Discussion

The aim of the present study was to longitudinally explore the role of two emotional regulation strategies (cognitive reappraisal and expressive suppression) and engagement in pleasant activities as protective and risk factors of affective and cognitive happiness during the Spanish COVID-19 lockdown. More specifically, we were interested in analyzing the effect of these emotion regulation strategies on the fluctuations in affective and cognitive happiness during the COVID-19 lockdown (in comparison with pre-pandemic levels) through the mediating role of engagement in pleasant activities during the lockdown.

Consistent with the results of previous studies where the COVID-19 lockdown has been demonstrated to have a significant psychological impact [6,7], the present study showed how cognitive happiness diminished during the lockdown in comparison with the period prior to the outbreak of the pandemic. Unexpectedly, no differences were found in affective happiness between the two timepoints. However, this finding is congruent with the results of a previous metanalysis showing that life events appear to have a greater impact on cognitive happiness than affective happiness [36]. Further, and in accordance with the Sustainable Happiness Model [15], the results of the present study revealed that those individuals that engaged more frequently in pleasant activities during the lockdown showed a smaller decrease in affective happiness. This provides support for H1 of the present study. In contrast, a similar correlation was not found with the cognitive happiness change (H2). However, we observed an indirect effect of engaging in pleasant activities on cognitive happiness, through the affective happiness change, which is consistent with the time-sequential model of well-being [11]. This issue will be discussed later.

In addition, and congruent with H3, those participants that employed the adaptative strategy of cognitive reappraisal appeared to more frequently engage in pleasant activities during the lockdown and had higher levels of affective (H4) and cognitive happiness (H5) both at timepoint 1 and timepoint 2. Previous studies have highlighted the relevance of this strategy as an adequate way of regulating emotions and achieving more happiness [17,19,20,21,22,23,24]. This result is also consistent with the findings reported in other studies conducted before the pandemic in which the cognitive reappraisal strategy was related to exercising more frequently [25]. With respect to the expressive suppression strategy, H6 and H8 were not supported, since the negative correlation between engaging in pleasant activities and cognitive happiness was not significant (although this approached significance). Future research should aim to provide a more in-depth analysis of how maladaptive strategies could impact both the extent to which individuals engage in these activities, and their cognitive happiness. Congruent with H7 and the results of previous studies—and contrary to the results found for the cognitive reappraisal strategy—expressive suppression was negatively correlated with affective happiness at timepoint 1 and timepoint 2. This finding was expected, given that expressive suppression is an emotional regulation ability that is usually regarded as inadequate [18,20,21,22].

Finally, and concerning the main focus of this study, a model has been offered for protecting people’s happiness during the COVID-19 lockdown period through the use of adaptative emotion regulation strategies and engagement in pleasant activities (H9). Specifically, the results showed how those participants that employed the cognitive reappraisal strategy engaged in more pleasant activities during the COVID-19 lockdown period. Subsequently, carrying out these pleasant activities during this period was related to higher levels of affective happiness, which, in turn, was related to higher levels of cognitive happiness, as predicted by the time-sequential model of well-being [11]. Thus, these results indicate that cognitive reappraisal could be an indirect protective factor of happiness in uncontrollable circumstances such as the COVID-19 lockdown, a finding that is consistent with the previous literature [17,18,19,20,21,22]. This makes sense, given that the aim of the reappraisal emotion strategy is to reevaluate a given situation, thus allowing people to look for alternatives to cope with the situation in a more adaptive way, which, in this case, was achieved by engaging in pleasant activities while confined at home [25]. Although the correlation between the expressive suppression strategy and engagement in pleasant activities did not reach significance (*p* = 0.07), this variable was included in the model as it slightly improves the fit of the proposed model.

Our results have important implications for the well-being of people during situations of confinement. From the three factors that influence happiness following the Sustainable Happiness Model [15], only one is in the hands of the individual, that is, intentional activities. In an uncontrollable situation such as that presented by the strict COVID-19 lockdown imposed in Spain where a reduction in the cognitive happiness of the participants was clearly evident, engaging in pleasant activities at home was one of the few things they could manage in order to improve their happiness. Tools such as the cognitive reappraisal strategy—which allows for increasing the frequency with which people engage in these activities [25]—could offer an adaptive way of improving people’s happiness during difficult times.

The present study has a number of strengths. First, this work focused on positive psychology, as opposed to psychopathology. As already mentioned, the previous literature has primarily focused on studying the latter, paying less attention to the protective factors that could help to address the emotional consequences of the pandemic and maintain happiness. This will be particularly useful for dealing with future lockdowns in an adaptive way. Second, this study employed a longitudinal design, evaluating how cognitive and affective happiness were altered during the COVID-19 lockdown by comparing the scores obtained during that period with those obtained prior to the pandemic. This allowed for measuring changes in happiness associated with the COVID-19 lockdown. Finally, our results have clear practical implications. The current limitations that people are experiencing as a result of COVID-19 are expected to continue for the foreseeable future [37], and it is, therefore, essential that people are able to draw upon resources that have the potential to improve their happiness. In this regard, the findings of this study suggest that promoting engagement in pleasant activities during periods of lockdown could help to mitigate some of the negative consequences of the pandemic. In order to achieve this, interventions aimed at improving happiness should focus on developing the most adaptive emotional regulation strategies, and according to the results reported here, training in the cognitive reappraisal strategy could be a first step towards achieving this goal. Therefore, brief interventions should focus on improving adaptive emotional regulation strategies such as emotional reappraisal [38]. In addition to this strategy, there are others (e.g., problem solving) that have also shown to be relevant for mental health [39]. The joint use of these strategies could help individuals to increase their happiness during these challenging times, although future lines of investigation should focus on empirically investigating the relevance of these alternative regulation strategies within the framework of the proposed model. Finally, together with an intervention based on emotional regulation strategies, it would be interesting to help individuals to identify the pleasant activities that are most enjoyable for them in order to facilitate the implementation of these strategies when necessary. Moreover, in the future, it might be interesting to identify what could be the most useful pleasant activities depending on sociodemographic and individual differences (e.g., people living alone, adolescents, marital status, or risk population).

Despite the strengths of this study, it is important to note some limitations. First, the size of the study sample was small. Only a small percentage of the participants involved in timepoint 1 decided to participate at timepoint 2. Here, it is important to note that when participants completed the assessment at timepoint 1, the second phase of the study had still not been planned, given that COVID-19 had not yet emerged. Thus, participants had not made any commitment to being involved in the second assessment. Although a larger number of participants would have helped to improve the representativeness of the sample, working with the present sample made it possible to employ a longitudinal design by taking advantage of data that had been collected prior to the pandemic, which has provided a unique way of examining the psychological consequences of COVID-19 and lockdown. Another limitation concerns the gender imbalance of the sample (with fewer men than women), which could potentially affect the generalizability of the findings. It should also be noted that the variance explained by the model was somewhat small. However, this is consistent with previous studies based on the Sustainable Happiness Model [16]. This model proposes that intentional activities explain only 15% of the variance in happiness. Therefore, in this study, the 8% of the variance explained by engagement in pleasant activities during the COVID-19 lockdown represents more than half of that predicted by the Sustainable Happiness Model. Finally, contrary to our expectations, no significant correlation was found between cognitive reappraisal and the change in cognitive and affective happiness. Nonetheless, this model revealed a positive indirect relationship between these variables through the frequency of engagement in pleasant activities.

## 5. Conclusions

In conclusion, the present study contributes towards understanding the mechanism(s) by which individuals can improve their happiness in stressful and restricted times such as the COVID-19 confinement. The theoretical implication is that happiness is not predetermined, but instead depends on adaptive emotional regulation strategies and intentional activities. In this work, a model is presented for maintaining happiness in potential future lockdown periods. The results of testing this model suggest that improving the cognitive reappraisal strategy could be an adaptive way of increasing the frequency with which people engage in pleasant activities during confinement, which will subsequently increase affective happiness and ultimately, cognitive happiness.

## Figures and Tables

**Figure 1 ijerph-18-03211-f001:**
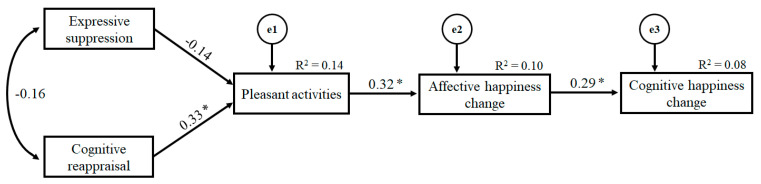
Graphical representation of the model including standardized path coefficients and explained variance (R^2^). An asterisk indicates significance at the *p* < 0.05 level.

**Table 1 ijerph-18-03211-t001:** Descriptive statistics (mean and standard deviation (SD)) of the total sample and separated by gender for the variables included in this study.

				Gender Comparisons	
	$\bar{x}$ (SD) Total Sample	$\bar{x}$ (SD) Men	$\bar{x}$ (SD) Women	*t*-Value	*p*-Value
Expressive suppression	3.40 (1.41)	3.52 (1.35)	3.35 (1.45)	0.53	0.60
Cognitive reappraisal	4.66 (1.18)	4.62 (1.32)	4.68 (1.12)	0.20	0.84
Pleasant activities	5.95 (1.25)	5.86 (1.13)	6.00 (1.31)	0.48	0.63
Affective happiness timepoint 1	0.82 (1.04)	0.86 (1.06)	0.80 (1.05)	0.22	0.83
Affective happiness timepoint 2	0.77 (1.19)	0.83 (1.33)	0.75 (1.13)	0.29	0.77
Affective happiness change	−0.05 (1.26)	−0.03 (1.10)	−0.05 (1.34)	0.09	0.93
Cognitive happiness timepoint 1	3.95 (1.08)	3.76 (1.27)	4.05 (0.97)	1.19	0.24
Cognitive happiness timepoint 2	3.53 (1.14)	3.59 (1.38)	3.51 (1.02)	0.30	0.77
Cognitive happiness change	−0.42 (1.00)	−0.17 (1.10)	−0.54 (0.93)	1.64	0.10

**Table 2 ijerph-18-03211-t002:** Pearson’s correlation matrix for the study variables.

	1	2	3	4	5	6	7	8
(1) Expressive suppression	-							
(2) Cognitive reappraisal	−0.16	-						
(3) Engagement in pleasant activities	−0.19	0.35 **	-					
(4) Affective happiness timepoint 1	−0.25 *	0.41 **	0.28 **	-				
(5) Affective happiness timepoint 2	−0.24 *	0.35 **	0.58 **	0.37 **	-			
(6) Affective happiness change	−0.02	−0.01	0.32 **	−0.48 **	0.64 **	-		
(7) Cognitive happiness timepoint 1	−0.19	0.42 **	0.40 **.	0.51 **	0.31 **	−0.12	-	
(8) Cognitive happiness timepoint 2	−0.19	0.27 *	0.51 **	0.38 **	0.48 **	0.14	0.60 **	-
(9) Cognitive happiness change	−0.00	−0.14	0.15	−0.11 **	0.20	0.29 **	−0.40 **	0.50 **

* *p* < 0.05, ** *p* < 0.01.

**Table 3 ijerph-18-03211-t003:** Standardized indirect effects and confidence intervals for the path model.

Path	Indirect Effect	Bias-Corrected 95% CI
Expressive suppression → Pleasant activities → Affective happiness change → Cognitive happiness change	−0.013	[−0.057, 0.002]
Cognitive reappraisal → Pleasant activities → Affective happiness change → Cognitive happiness change	0.030 *	[0.004, 0.080]
Pleasant activities → Affective happiness change → Cognitive happiness change	0.092 *	[0.011, 0.206]

Note: * significant indirect effect.

## Data Availability

Data are available upon reasonable request by emailing M.J.G.-C.

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
