# Peer review of "Is It Possible to Be Happy during the COVID-19 Lockdown? A Longitudinal Study of the Role of Emotional Regulation Strategies and Pleasant Activities in Happiness"

_ijerph, 2021, doi:10.3390/ijerph18063211_

Round 1
Reviewer 1 Report
- This manuscript aims to to longitudinally analyze the role played by two emotional regulation strategies (cognitive reappraisal and expressive suppression), through the mediating effect of engagement in pleasant activities during lockdown, in changes in affective and cognitive happiness in comparison with pre-pandemic levels. It deals with a ver important topic: the impact of lockdown on mental health.
- My greatest concern is with the contribution of the present research. Authors should more explicitly lay out the importance of the research to the readers.
- It is not clear to me whether you draw your study in Sustainable Happiness Model. If so, please explicitly indicate in what ways this model fits your study and if you are using this model as the theoretical framework, authors should describe this theory in a more detailed way.
- Did you use any control variables? For example, I do believe that the marital status or having children do affect the results.
- Hypotheses: I think you should split your hypotheses, so you have one hypothesis for affective happiness and other for cognitive happiness.
- You examined gender and age-related differences in the study variables. Why? I believe that your hypotheses do not include gender and/or age-related differences.
- How do you explain the non-significant differences in affective happiness between Timepoint 1 and Timepoint 2?
- According to your theoretical model, affective happiness in antecedent of cognitive happiness. Why? Please explain this path providing some theoretical framework.
- Please propose some explanation for the non-significant results.
- What are the theoretical implications of the research?
- The implications for practice should be more developed and clearly stated.
Reviewer 2 Report
Interesting study, where the results have important implications for healthcare services during circumstances like the current pandemic. I have some suggestions for how to improve the manuscript.
Abstract:
Please rephrase the sentence “In turn, these pleasant activities were related to more affective happiness during the lockdown (compared with pre-pandemic levels), which was associated with greater cognitive happiness”. What does “which” refer to (I suspect it refers to affective happiness, but this could be clarified)? The last sentence may be removed (unnecessary).
Introduction:
It is stated that “Cognitive reappraisal, in comparison with expressive suppression, has previously been linked to better mental health and — of particular relevance for the purposes of the present study — to happier individuals [17–22].” Please elaborate a bit on the findings our prior studies. How, specifically, have the two strategies been linked with mental health and happiness? As I interpret the statement, expressive suppression has been found to be less clearly associated with desired outcomes. If this is correct, I believe the authors should also provide a rationale for including expressive suppression in their model.
Methods:
Please clarify the response options for the question concerning pleasant activities.
The community sample recruited at T1 consisted of 783 persons. How well does the current sample (n=88) reflect the characteristics of the sample at T1? The attrition analyses might be used as indicative of the sample’s ability to represent the study population.
The authors state that “Second, we examined gender and age-related differences in the study variables using independent two-sample t-tests and Pearson’s correlations, respectively. Moreover, changes in levels of affective and cognitive happiness between Timepoint 1 and Timepoint 2 were analyzed by independent paired sample t-tests.” I believe the use of correlation analyses here requires a linear relationship (e.g., to age) to be meaningful. Are the variables’ relationships to age linear? The expression ‘independent paired sample t-tests’ is a contradiction in terms. Paired samples imply that the comparisons are performed within individuals across two time points, that is, the data points are dependent as they come from the same individuals.
Please include a statement on statistical significance – what threshold is decided upon? Refer to this threshold consistently throughout the Results section. Reading about ‘marginally significant results (p =0.07)’ in the Results is puzzling.
Results:
The authors state that “Gender differences were not observed for any variable (all ps < .05)”. If the text statement is correct, p should be > .05. In Table 1, please include a column displaying the p values related to the gender comparisons.
They also state that “Age was only found to be related to the variable of affective happiness at Timepoint 1 (r = .27, p = .01)”. Please write in a form that reveals the direction of the detected association. I assume this should be that higher age was related to higher affective happiness at T1.
I have informed the Editor that I am not fully able to assess the validity of the mediation analysis. However, the presentation is logical and easy to follow.
Discussion:
While this section discusses the main points in coherent fashion, it would improve by incorporating more references throughout. When discussing the mediation model, only two references are used (one of which is a reference to the theoretical model). More context is needed.
Relevant limitations of the study are mentioned. However, it might also be useful to suggest how future studies can build from this one to improve the state of knowledge.
Minor points:
Some sentences are awkwardly phrased; a language check would be helpful (e.g., line 319-320 “in particular there were few men in comparison with women, makes between-group comparisons difficult”).
Line 42: COVID-19
Line 46 (and elsewhere): the personal writing style (“protective factors of our happiness”) appears misplaced. Please rephrase.
Line 267: “…expressive suppression was negatively correlated with affective happiness, which is an ability that is usually regarded as inadequate”. It is unclear what ability is referred to; please rephrase the sentence.
Line 299: “We evaluated how cognitive and affective happiness were altered as a consequence of the COVID-19 lockdown”. While this sounds intuitive, it is impossible to state cause and effect, based on the presented data. Please rephrase.
Some irregularities in the reference list.
Reviewer 3 Report
- There seems to be a huge difference in the number of participants between timepoint 1 and time point 2 -- I think commenting on this will be helpful (why, or do you think there was a selection bias of some sort, etc...).
- Using "differential" to indicate change/gain is a bit confusing -- perhaps say "change?"
- Conclusion can be improved.
Author Response
Please see the attachment

This manuscript is a resubmission of an earlier submission. The following is a list of the peer review reports and author responses from that submission.